# Congenital Hyperinsulinaemic Hypoglycaemia—A Review and Case Presentation

**DOI:** 10.3390/jcm11206020

**Published:** 2022-10-12

**Authors:** Sylwia Krawczyk, Karolina Urbanska, Natalia Biel, Michal Jakub Bielak, Agata Tarkowska, Robert Piekarski, Andrzej Igor Prokurat, Malgorzata Pacholska, Iwona Ben-Skowronek

**Affiliations:** 1Department of Paediatric Endocrinology and Diabetology, Medical University of Lublin, 20-093 Lublin, Poland; 2Department of Neonate and Infant Pathology, Medical University of Lublin, 20-093 Lublin, Poland; 3Department of Paediatric Surgery, Regional Children’s Hospital in Bydgoszcz, 85-667 Bydgoszcz, Poland

**Keywords:** congenital hyperinsulinaemic hypoglycaemia, congenital hyperinsulinism, ABCC8, neonatal hypoglycaemia

## Abstract

Hyperinsulinaemic hypoglycaemia (HH) is the most common cause of persistent hypoglycaemia in infants and children with incidence estimated at 1 per 50,000 live births. Congenital hyperinsulinism (CHI) is symptomatic mostly in early infancy and the neonatal period. Symptoms range from ones that are unspecific, such as poor feeding, lethargy, irritability, apnoea and hypothermia, to more serious symptoms, such as seizures and coma. During clinical examination, newborns present cardiomyopathy and hepatomegaly. The diagnosis of CHI is based on plasma glucose levels <54 mg/dL with detectable serum insulin and C-peptide, accompanied by suppressed or low serum ketone bodies and free fatty acids. The gold standard in determining the form of HH is fluorine-18-dihydroxyphenyloalanine PET ((18)F-DOPA PET). The first-line treatment of CHI is diazoxide, although patients with homozygous or compound heterozygous recessive mutations responsible for diffuse forms of CHI remain resistant to this therapy. The second-line drug is the somatostatin analogue octreotide. Other therapeutic options include lanreotide, glucagon, acarbose, sirolimus and everolimus. Surgery is required in cases unresponsive to pharmacological treatment. Focal lesionectomy or near-total pancreatectomy is performed in focal and diffuse forms of CHI, respectively. To prove how difficult the diagnosis and management of CHI is, we present a case of a patient admitted to our hospital.

## 1. Introduction

Hyperinsulinaemic hypoglycaemia (HH) is the most common cause of persistent hypoglycaemia in infants and children [1]. It is characterised by dysregulated insulin secretion from pancreatic β-cells despite low glucose levels [2]. It can be either congenital or acquired. Congenital hyperinsulinism mutations in genes related with the regulation of insulin secretion have been described. These genes are ABCC8, KCNJ11, GLUD1, GCK, HADH, UCP2, SLC16A1, HNF4A, HNF1A, HK1, PGM1 and PMM2 (Table 1) [3,4]. Congenital forms of hyperinsulinism (CHI) can be divided into the following two main groups: channelopathies defects in ion channels located on β-cells membrane and metabolopathies defects in the metabolic pathways leading to increased insulin release. The most common mutations are those affecting the KATP channel genes–ABCC8 (36.8%) and KCNJ11 (5.9%) (Figure 1) [5]. Some of the cases of HH are associated with genetic syndromes; Beckwith–Wiedemann syndrome is the most common [6]. The incidence of CHI is estimated at 1:50,000 live births, but in some populations, such as Finland or Saudi Arabia, the incidence reaches from 1:2500 to 1:3000 live births [7]. Transient forms of HH are secondary to conditions such as perinatal asphyxia, intra-uterine growth retardation, erythroblastosis fetalis, maternal diabetes mellitus or the administration of intravenous glucose during labor [6].

## 2. Pathophysiology and Symptoms of CHI

The K_ATP_ channel reacts to an increased concentration of ATP (caused by glucose metabolism) by closure of the channel. This leads to β-cell membrane depolarisation and opening of the voltage-dependent Ca^2+^ channel. The increase in intracellular Ca^2+^ triggers insulin secretion [8]. The K_ATP_ channel consists of four sulphonylurea receptor 1 (SUR1) subunits, encoded by ABCC8, and four inward-rectifier potassium channel (Kir6.2) subunits, encoded by KCNJ11 [9]. The most severe forms of HH are caused by an inactivating mutation of the ABCC9 and KCNJ11 genes, which together are responsible for 36–70% of CHI cases [10,11].

Insulin decreases blood glucose concentrations by driving it into the insulin-sensitive tissues and inhibiting gluconeogenesis and glycolysis [2]. Insulin also inhibits lipolysis and ketone body synthesis. Therefore, in hyperinsulinaemic hypoglycaemia, the brain cannot be supplied with alternative energy sources. The combination of hypoketonaemic hypoglycaemia may lead to brain damage [5]. CHI is symptomatic mostly in early infancy and the neonatal period. Severe hypoglycaemia may even cause seizures or/and coma. The newborns are usually too big for their gestational age as a consequence of the high insulin level acting as a growth factor in utero [11]. Less specific symptoms of hypoglycaemia in the neonatal period include poor feeding, lethargy, irritability, apnoea and hypothermia [2]. In clinical examination, some newborns present cardiomyopathy and hepatomegaly [9].

## 3. Diagnosis of HH

An infant’s glucose blood concentration at birth amounts to 70% of the maternal levels. Within 1 h, it may decrease to 20–25 mg/dL. Then, over the next hours and days, it begins to rise [12]. The diagnostic criteria for patients with HH are based on the clinical presentation and biochemical profile. Plasma glucose levels of less than 54 mg/dL are accompanied by detectable serum insulin, detectable C-peptide, suppressed or low serum ketone bodies and suppressed or low serum concentrations of free fatty acids. The supportive evidence is a glucose infusion rate of >8 mg/kg per minute which is required to maintain normoglycaemia [9]. In the case of an unclear cause of hypoglycaemia, a glucagon test is performed. In this test, glucagon is administered intravenously at a dose of 0.03 mg/kg, and then blood glucose is measured at 0, 10, 20 and 30 min. CHI is diagnosed when the blood glucose level is higher than 30 mg/dL [9].

Histologically, CHI is divided into three types–diffuse, focal and atypical [13]. Focal lesions are responsible for 30–40% of congenital hyperinsulinaemic hypoglycaemia (CHH). Abnormal pancreatic cells are limited to a single location. The diffuse form of CHH accounts for 60–70% cases. The pathological β-cells are disseminated throughout the pancreas. The atypical form is described as a mosaic pattern of the two previous forms [6]. The gold standard in determining the form of HH is fluorine-18-dihydroxyphenyloalanine positron emission tomography ((18)F-DOPA PET). The sensitivity and specificity of (18)F-DOPA PET or PET/CT in differentiating between focal and diffuse CHI is 89% and 98%, respectively [14]. Rapid genetic testing of K_ATP_ channel genes allows for determining the histological subtype of the disease. It should be considered in diazoxide-unresponsive cases before performing (18)F-DOPA PET. The biallelic or single dominant K_ATP_ channel disease-causing variant confirms the diffuse form of HH, while a paternally inherited recessive K_ATP_ channel variant is linked to the focal form with a sensitivity of 97% [15,16,17].

### Differential Diagnosis

The causes of neonatal hypoglycemia can be divided into four main groups: hyperinsulinism, inborn metabolic errors, counter-regulatory hormone deficiency and increased glucose requirement (Table 2) [18].

The majority of neonatal hypoglycemia cases are caused by delayed metabolic adaptation after birth. Some patients require high glucose infusions for a week or more until they develop physiological metabolic processes [19].

Metabolic or hormonal etiology should be suspected in low-risk cases or unusual severity of hypoglycemia [19]. Most of these causes can be excluded or confirmed with laboratory testing [19,20,21,22,23] (Table 3). The critical samples for laboratory investigations should be obtained at the time of a hypoglycaemic episode [19].

## 4. Treatment

The aim of the therapy in CHI is to:-Stabilise blood glucose levels at >60 mg/dL;-Prevent neuroglycopaenia (manifested in hypothermia, apnoea, feeding problems, Cyanosis, tremor, convulsions, apathy, and coma);-Prevent long-term neurological problems (such as epilepsy, physical and psychological developmental delay, and microcephaly).

### 4.1. Diazoxide

The first-line treatment for CHI is diazoxide, which binds to the linker region of the SUR1 subunit of the K_ATP_ channels and suppresses β-cell insulin secretion by forcing the channels to remain open [6,24,25,26,27,28]. Recent studies suggest diazoxide as the drug of choice and recommend a starting dose of 5 mg/kg/day with target values of no more than 20 mg/kg/day [6,25,27,28]. One study reported that high diazoxide doses of >5 mg/kg/day, necessary to maintain euglycaemia, can be considered a severity and prognostic marker in the early stages of diagnosis [29]. Dominant and recessive inactivating mutations in the ABCC8 gene are, on the other hand, an important predictive factor. Homozygous or compound heterozygous recessive mutations, responsible for the diffuse form of CHI, are unresponsive to diazoxide and usually require subtotal pancreatectomy to achieve euglycaemia [3,6,27,30]. Recessive K_ATP_ channel mutations can also cause focal diazoxide-unresponsive CHI, when a paternally-transmitted mutation becomes isodisomic due to somatic loss of the maternal 11p imprinted region [3,6,30]. Dominant mutations are less common, but are associated with all K_ATP_-CHI cases with diazoxide-responsive CHI, and about one-third of diazoxide-unresponsive diffuse forms [3,30]. Diazoxide unresponsiveness is often thought to be a hallmark of recessively inherited ABCC8 mutations. Partial diazoxide responsiveness, in combination with octreotide treatment, may be seen in children with homozygous ABCC8 gene mutations, which may prevent more aggressive treatment with numerous side effects [31]. In view of the possible side effects, the optimal dose of diazoxide should be as low as possible to maintain euglycaemia [27]. Diazoxide hypersensitivity, which manifests as hyperglycaemia at minimal doses of diazoxide, is very rare and is usually associated with a mutation in the HNF4A gene [25]. Hypertrichosis is the most common adverse effect of diazoxide and its degree is not correlated with the dose of the drug [28,32]. Fluid retention and heart failure are also common symptoms [27,28,32]. Diazoxide used to be a hypotensive drug as it acted as a vasodilator. However, diazoxide causes sodium and chloride retention in the renal tubules and decreases bicarbonate excretion and free water clearance, leading to fluid retention [33,34]. Therefore, diazoxide should be used with caution in CHI patients who require high-volume intravenous or oral fluid therapy to maintain normoglycaemia. Fluid restriction is therefore advisable, along with the administration of chlorothiazide (which also has a synergistic action over the K_ATP_ channels [6,27,35]), or furosemide with spironolactone to prevent fluid overload [31]. In extreme preterm infants and children with lung disease and congenital heart defects, diazoxide can induce reversible pulmonary hypertension (caused probably by direct cardiac toxicity [36]), reopening of the ductus arteriosus, sepsis syndromes, heart failure, neutropenia, thrombocytopenia, hyperuricemia, and hyperosmolar coma [24,26,27,35,37,38].

### 4.2. Octreotide

A second-line drug is the somatostatin analogue octreotide [6,39,40], whose mechanism involves binding to somatostatin receptors 2 and 5 (SSTR2 and SSTR5). SSTR5 activation reduces the insulin gene promoter activity, inhibits calcium mobilisation and acetylcholine activity. Somatostatin also decreases insulin secretion by the activation of K_ATP_ channels and G protein-regulated inwardly rectifying K^+^ channels, which act by decreasing voltage-gated Ca^2+^ influx and inhibiting insulin exocytosis [41]. The starting dose is 5 μg/kg/day, administered subcutaneously or as a continuous infusion at 6 to 8 h intervals, with a maximum dose of 30–35 μg/kg/day. Continuous subcutaneous infusion with an insulin pump is an alternative to surgery in patients with monoallelic K_ATP_ channel mutations [39,40]. The initial response to octreotide administration is usually hyperglycemia, followed by a blunted effect after 48 h (tachyphylaxis). Numerous studies confirm that it is an effective treatment in cases unresponsive to diazoxide [27,42]. Octreotide binds to SSTR2 and SSTR5; prolonged use, however, may develop into drug desensitization due to an internalization of the receptors [6]. Adverse effects of octreotide include pain, stinging and burning sensations, redness and swelling at the injection site, in addition to delayed gastric emptying, decreased gallbladder contractility, nausea, abdominal cramps, diarrhoea, fat malabsorption, and flatulence. Somatostatin analogues inhibit the secretion of many hormones, including growth hormone, without affecting growth with long-term use. The most common sequelae of long-term octreotide therapy include the transient elevation of liver enzymes and asymptomatic cholecystolithiasis. Octreotide decreases portal blood flow and should therefore be used with caution in infants at risk of necrotizing enterocolitis [40]. It is advisable to test the serum concentration of the drug in order to choose the optimal dose that provides an adequate therapeutic effect while minimizing the risk of adverse effects [43].

### 4.3. Lanreotide

The sustained-release octreotide formulation and lanreotide are both long-acting somatostatin analogues that are administered once every 28 days, which increases the treatment adherence and improves Quality of Life (QoL) [44]. They have been shown to provide more stable glycaemic control than conventional ocreotide formulations [45]. Additionally, lanreotide may provide favorable treatment outcomes in inoperable focal CHH [46].

### 4.4. Glucagon

Glucagon can also be used in the treatment of HI [6,39]. It is administered intravenously or subcutaneously [2]; however, the subcutaneous route may be difficult to handle due to catheter obstruction occurring every 2 to 3 days, or even daily [47,48]. Glucagon helps to stabilize glycemic levels during initial treatment or before surgery and, in combination with octreotide or alone, can be used in the acute management and prevention of near-total pancreatectomy in infants with HI [48]. Additionally, ketone production can be restored by enhancing the glucagon signaling pathway sufficiently enough to overcome the insulin inhibitory effect on ketogenesis [47].

### 4.5. Acarbose

Acarbose is used in the treatment of postprandial HI when a proper diet fails to prevent glycaemic decline. It works by slowing glucose absorption, so that there is no glycemic peak followed by insulin release [49].

### 4.6. Mammalian Target of Rapamycin (mTOR) Inhibitors

One of the new treatment methods is the use of mTOR inhibitors such as sirolimus and everolimus. The intracellular mTOR pathway is involved in β-cell growth and altered insulin secretion in patients with insulinoma, and ketone body synthesis [27,50,51,52]. The mechanism of action of sirolimus in the treatment of HH is not well understood. Sirolimus provides good glycaemic control, which makes it possible to avoid pancreatectomy. The response to sirolimus is independent of the genetic aetiology of HH. The most commonly reported adverse effects are stomatitis, increased risk of infection, immunosuppression, renal dysfunction, fatigue, pneumonitis and increased serum aminotransferase or lipid levels [6,50,53]. A few individual cases of sirolimus-induced hepatitis have also been described [52].

### 4.7. Potential New Therapies

Potential new therapies for the future treatment of patients with HH include glucagon-like peptide-1 (GLP-1) antagonists, pharmacological trafficking chaperones, such as carbamazepine, sulphonylurea glibenclamide and tolbutamide, and stable glucagon formulation [6,27].

## 5. Dietary Treatment

Dietary treatment consists of frequent feeding, long-acting carbohydrates, abundant protein, fiber supplements, and fat emulsions. This management aims to prevent a sudden rise in blood glucose that would cause a rapid insulin release and subsequent hypoglycemia [54]. In addition, the acute hypoketotic neuroglycopenia that occurs in HH and gluconeogenesis inhibition make the brain more susceptible to the neurological manifestations of hypoglycemia. There is a study suggesting that the use of a ketogenic diet may have a neuroprotective effect, which manifests itself in some patients as freedom from epileptic seizures, improved mental development and higher QoL [27].

## 6. Surgical Treatment

Acute hypoglycaemic episodes that cannot be eliminated with the use of maximum doses of medication and require continuous intravenous glucose infusions are an indication for surgical treatment, especially with the focal form of the disease (Figure 2).

Histologically, CHI is divided into limited and diffuse forms, the differentiation of which is important in planning surgical treatment [6,27,30,39,55]. It is important to distinguish between the limited and diffuse forms, as in the former case surgical removal of the lesion offers the possibility of complete cure for the patient [6,29,55]. Its forms are rarely hereditary and are usually located in the tail and body, but can also be found in any other part of the pancreas. Sometimes it is impossible to localise the tumour using contrast CT or MRI, especially in the case of small insulinomas [56]. Endoscopic Ultrasonography has been proven to be an effective method of visualisation in these situations with a sensitivity of 83.3% for tumours located in the head of the pancreas. However, in the case of tumours located in the tail, sensitivity is lower (37%) [57]. Ultrasonography is very useful both in localizing the tumour and delineating the anatomy of related structures [58]. Some experts recommend intraoperative ultrasound for localising the functioning tumour directly for pancreatic exploration [59]. If the pre-operative imaging studies are negative, the intra-arterial calcium stimulation test and assessment of insulin gradient in the gastroduodenal artery is recommended for confirming the regionalisation of insulin excess [60,61]. The diagnostic algorithm suggests verifying the location by performing intraoperative ultrasound and multiple biopsies investigated with histopathologic methods. If the location is not reached, a pancreatic tail resection for further histologic evaluation is recommended [60,61]. Treatment consists of the excision of the focal lesion [6,39,55]. Laparoscopic removal is the method of choice when the lesion is located in the body or tail. When the location is in the head of the pancreas, laparotomy is performed. During surgery, it is important to perform an intraoperative biopsy to ensure complete excision with histologically confirmed margins [6].

In the diffuse form, the lesions are dispersed throughout the pancreas. Pharmacological treatment is usually insufficient and subtotal pancreatectomy (involving nearly 95–98% of the pancreas) is required for the normalization of glycaemia. Laparoscopy is currently the preferred method [6,27,39,55,62]. Unfortunately, this form of treatment has unsatisfactory results in most cases [6]. The possible complications of near-total pancreatectomy include ongoing CHI, diabetes mellitus, pancreatic exocrine insufficiency and damage to the common bile duct [6,31,39]. Sometimes, despite performing subtotal pancreatectomy, symptoms of HI still persist, along with postprandial hyperglycaemia [27]. Despite the above adverse effects, it is still a beneficial method as it is usually easier to manage with dietetic/medical therapy [6].

## 7. Influence of CHH on Postnatal Development and QoL

In the literature, there is a lack of long-term large cohort studies investigating the growth and weight of patients with HI. There are many articles reporting high body weight at birth, but only a few focusing on later postnatal development, which usually describe it as normal, without providing any precise data [29]. A significant proportion of patients with HI show feeding difficulties which start early in life [63] and may persist for many years [64]. These are often caused by breast- and bottle-feeding problems, diazoxide-induced nausea, or supra-physiological caloric intake to maintain euglycemia through enteral and intravenous dextrose administration [65]. A significant proportion of children require antireflux treatment, nasogastric tube feeding or gastrostomy; this proportion, however, decreases with disease duration [63,65]. Patients with feeding problems require intravenous glucagon infusions due to severe hypoglycemia [62]. Additionally, feeding problems are more common in patients with the diffuse form of HI undergoing subtotal pancreatectomy than in patients with the limited form treated with lesionectomy, as feeding problems decrease approximately 6 months after diagnosis [63,64].

Neurodevelopmental problems are also common in CHI patients and their frequency does not differ between transient and permanent forms. Their severity is significantly higher in patients with a more severe course of CHI, as determined by maximum doses of diazoxide and early symptoms at <7 days of age [66]. Neurodevelopmental problems may include sucking and swallowing disorders [63], speech, language, motor and vision delay, seizures, infantile spasms, lower limb weakness, learning problems during the school years, higher risk of epilepsy in later childhood, and cerebral palsy [28,55,65]. These problems result from hypoketotic neuroglycopenia, which causes damage to the central nervous system. However, despite the presence of abnormal neurodevelopmental outcomes, it is not always possible to visualise these abnormalities on neuroimaging studies [39,66].

Health-related quality of life (HRQoL) in CHI was examined using the general KINDL-R questionnaire collected from children and their parents. Interestingly, it showed that, despite the numerous health problems, CHI is not associated with lower HRQoL in childhood or adolescence [67].

There are reports that heterozygous patients with an inactivating mutation in the ABCC8 gene responsible for the mild form of HH (even those with no history of pancreatectomy) occasionally develop diabetes later in life, and that their family members often have diabetes, which can be successfully treated with sulphonylurea [62,68,69,70,71,72,73]. One study performed on transgenic mice with inactivating K_ATP_ channel mutations showed that unresponsiveness to glucose, increased apoptosis and/or decreased insulin gene expression in the pancreatic β-cells might be associated with the development of diabetes caused by the inactivating mutations [68]. Another study established that the same mechanism is present in humans and that diabetes of this etiology responds well to sulphonylurea derivatives [73]. This indicates the need for long-term follow-up in patients with the ABCC8 gene mutations in view of the increased risk of diabetes later in life [62], and for a multidisciplinary approach due to multiple health problems from the neonatal period onwards [63]. Such children should remain under the care of a pediatric endocrinologist, nutritionist, surgeons, developmental pediatricians and psychologists. They require regular control of serum drug levels, blood glucose monitoring and clinical observation towards potential adverse effects. Post-surgical patients require periodic follow-ups with laboratory testing, including stool elastase, capillary glucose, and an oral glucose tolerance test, in search of complications, such as hypoglycemia, diabetes, and exocrine pancreatic insufficiency. A large number of children with HH have been diagnosed with various neurological disorders, epilepsy, and neurodevelopmental delay which may require a special education plan devised with the help of an educational psychologist and developmental pediatrician [6].

## 8. Case Report

A child from the third pregnancy was born in the 37th week of gestation by caesarean section, which was performed on account of severe intrauterine asphyxia (the mother did not feel fetal movements for 12 h). The newborn presented macrosomia (body weight—5500 g, body length—65 cm, head circumference—37 cm, breast circumference—39 cm). Due to the 1 min Apgar score of 2, he required intubation, resuscitation and further treatment in the Intensive Care Unit. In view of very low serum glucose levels (21–31 mg/dL), continuous intravenous 12.5% glucose infusion was provided. During the first hours of life, the patient was diagnosed with pneumonia and treated with antibiotics. He also presented decreased pain reaction, increased muscle tone and asymmetrical Moro reflex. Despite glucose infusion, very low levels of serum glucose (7 mg/dL) were observed. A total of 20% glucose solution was administered with a central catheter. Echocardiography performed on the first day of life revealed massive cardiomegaly with right ventricular and septal hypertrophy (without left ventricular outflow tract obstruction), patent ductus arteriosus with left-to-right shunt, mild mitral and tricuspid insufficiency, and elevated right ventricular pressure. In an abdominal ultrasound, no abnormalities were found. Cranial ultrasound and brain MRI showed numerous ischemic foci within the white matter of the brain and cavum septum pellucidum.

Due to hypoglycaemia, the patient required continuous glucose infusion of >15 mg/kg/min. The extremely high serum insulin levels (up to 192 µIU/mL) were observed in relation to the episodes of hypoglycaemia. During these episodes, the counterregulatory hormones (growth hormone, ACTH and cortisol) remained elevated; however, as they were secreted adequately to the low glucose levels, hypopituitarism was excluded. In the first week of life, the newborn was also treated with hydrocortisone as congenital adrenal hyperplasia (CAH) was suspected in view of scrotal hyperpigmentation. However, 17-OH progesterone levels remained normal; therefore, the hydrocortisone dose was gradually reduced. The patient received both parenteral and nasogastric tube feeding. The tandem mass spectrometry (MS/MS) screening tests for inborn errors of metabolism as a potential cause of hypoglycaemia were negative. Congenital adrenal hyperplasia was also excluded based on steroid profiling of the 24 h urine sample (Table 4).

Beckwith–Wiedemann and Sotos syndromes were excluded using the methylation-specific multiplex ligation-dependent probe amplification (MS-MLPA) method. On the 42nd day of life, however, the results of the whole exome sequencing (WES) analysis revealed two abnormalities in the ABCC8 gene related to familial hyperinsulinism: c.3989-9G>A and p.Ala113Val. The c.3989-9G>A variant is a known pathogenic mutation responsible for persistent hyperinsulinaemic hypoglycaemia of infancy (PHHI). This diagnosis was confirmed with the Sanger method.

Due to recurrent episodes of hyperinsulinaemic hypoglycemia, the patient required continuous intravenous glucose infusion (20–40% glucose at the rate of 50–20 mL/kg/h), as well as nasogastric tube feeding. He also had trouble with coordination of sucking and swallowing. Because of the poor treatment results, diazoxide therapy was introduced 18 days after birth, at a starting dose of 7 mg/kg/day. It led to the reduction of the 20% glucose dose to 8 mL/kg/h. However, the adverse effects of diazoxide, namely hypertrichosis, hyperuricemia and dyspepsia, were observed. Octreotide was added to the therapy at a dose of 32 μg/kg/day. Unfortunately, episodes of hypoglycemia of lower than 40 mg/dL were observed despite the continuous glucose infusion. The decision of surgical treatment was made.

When the patient was 6 months old, he was admitted to the Department of Pediatric Surgery. F-DOPA PET confirmed a diffuse form of PHHI (Figure 3). The boy was qualified for laparotomy with subtotal pancreatectomy. During the surgery, 96% of the patient’s pancreas was removed.

After subtotal pancreatectomy, the diazoxide and octreotide therapy was discontinued. The infant was provided with parenteral feeding in the first week following the surgery. At this time, he presented a significant tendency towards hyperglycaemia (160–180 mg/dL). After the feeding method was changed from nasogastric to enteral, there were a few incidents of mild hypoglycaemia (with the minimal glucose level of 50 mg/dL).

The patient was transported to the Department of Paediatric Endocrinology and Diabetology for further treatment. In the next 3 weeks, he was placed under the care of a neurologist and speech therapist because of the problems with oral feeding. The oral diet was widened stepwise. The boy learned to eat with a spoon. The parents were advised to stay in contact with a dietician because the patient required a balanced diet of carbohydrates, proteins and fats, as well as proper breaks between meals. Dinner and the last meal before sleep were especially important for fear of nocturnal hypoglycaemia. No severe neuroglycopaenia symptoms were observed. During the first month of treatment, the boy was under continuous glucose monitoring (CGM), which was later replaced with flash glucose monitoring (FGM). Serum glucose, HbA1C, C-peptide, lipase and amylase, as well as faecal elastase, were checked in the follow-ups every 2 months (Figure 4).

The cardiac hypertrophy remained at the same level. As the thickness of the left ventricle increased ever since, the discrepancy between the norm and the patient’s left ventricle decreased.

The patient remains under the multidisciplinary specialist care of an endocrinologist and diabetologist, as well as gastrologist, cardiologist, neurologist, physiotherapist, speech therapist, psychologist and dietician. The boy’s motor and intellectual development is delayed 6 months but correct. Serum glucose levels remain normal most of the time, just as the parameters of the exocrine and endocrine functions of the pancreas (Table 5).

To summarise, persistent non-focal CHI remains difficult to manage. A cohort study described by Rasmussen et al. showed neurological impairment of 30% and problematic status upon follow-up of 18% patients. It suggests the lack of prompt and adequate treatment and the urgent need to improve early diagnosis, treatment and new medical treatment modalities [74].

## 9. Conclusions

Even though CHI is quite rare, it may be a life-threatening condition, especially in early infancy. Diagnosis is not easy as CHI is associated with many inherited mutations and a few genetic syndromes. The gold standard diagnostic method (18)F-DOPA PET is not widely accessible in hospitals. Some patients remain unresponsive to pharmacological treatment and require surgery. Unfortunately, even radical surgical treatment does not guarantee a complete elimination of sporadic hypoglycaemias. Nevertheless, although diagnosis and treatment is difficult and complex, it is still possible for the patients to run relatively normal lives.

## Figures and Tables

**Figure 1 jcm-11-06020-f001:**
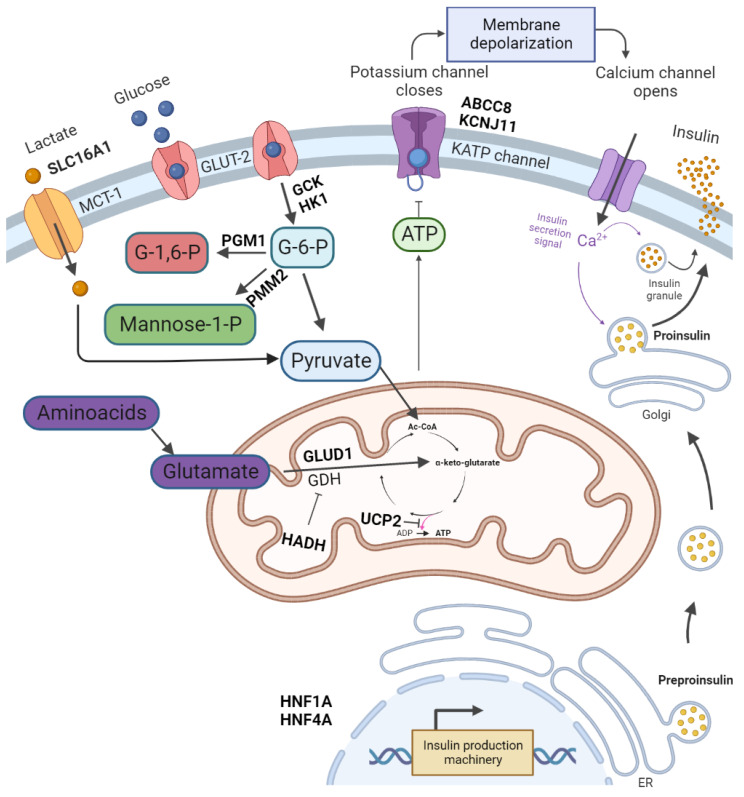
The regulation of insulin release and gene mutations involved in CHI pathogenesis. Glucose enters pancreatic β-cell via glucose transporter 2 (GLUT-2). It is then phosphorylated by glucokinase. The product of glycolysis–pyruvate is a substrate of the tricarboxylic acid cycle in mitochondria. The tricarboxylic acid cycle product is ATP. An increase in ATP levels leads to a sequence of K_ATP_ channel closure, membrane depolarisation, opening of voltage-dependent Ca^2+^ channels, Ca^2+^ influx and eventually insulin exocytosis. Glutamate dehydrogenase (GDH) catalyses the oxidative deamination of glutamate to alpha-ketoglutarate and ammonia. Further processes lead to amino acid-induced insulin secretion. PGM1 and PMM2 are responsible for glycosylation. Monocarboxylate transporter 1 (MCT1) catalyses the transport of monocarboxylates–pyruvate and lactate. Enhanced MCT1 expression increases pyruvate metabolism in the tricarboxylic acid cycle. Created with BioRender.com, accessed on 22 September 2022.

**Figure 2 jcm-11-06020-f002:**
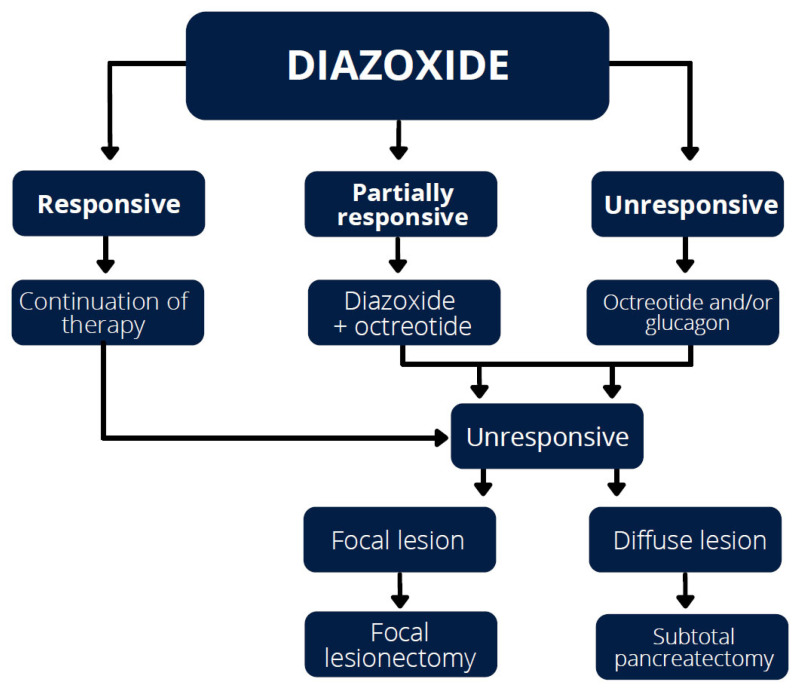
Treatment of HH.

**Figure 3 jcm-11-06020-f003:**
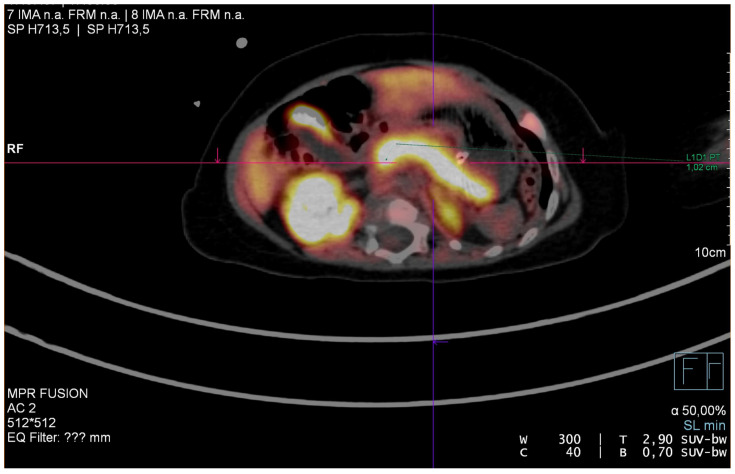
Diffuse form of CHI in F-DOPA PET of our patient.

**Figure 4 jcm-11-06020-f004:**
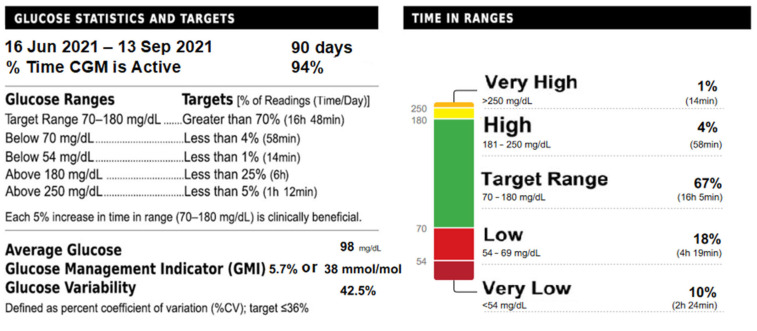
90-day Flash Glucose Monitoring Report.

**Table 1 jcm-11-06020-t001:** Genes in CHI pathogenesis.

Gene	Full Gene Name	Diazoxide Responsiveness
ABCC8	ATP-Binding Cassette Subfamily C Member 8	Yes/No
KCNJ11	Potassium Inwardly-Rectifying Channel Subfamily J Member 11	No
GLUD1	Glutamate Dehydrogenase 1	Yes
GCK	Glucokinase	Yes/No
HADH	Hydroxyacyl-CoA Dehydrogenase	Yes
SLC16A1 (MCT-1)	Solute Carrier Family 16 Member 1 (Monocarboxylate Transporter Subtype 1)	Yes/No
UCP2	Uncoupling Protein 2	Yes
HNF4A	Hepatocyte Nuclear Factor 4A	Yes
HNF1A	Hepatocyte Nuclear Factor 1A	Yes
HK1	Hexokinase 1	Yes
PGM1	Phosphoglucomutase 1	No
PMM2	Phosphomannomutase 2	Yes

**Table 2 jcm-11-06020-t002:** Main causes of neonatal hypoglycaemia.

Transient neonatal hypoglycemia caused by hyperinsulinism	Infant of diabetic mother Decreased pancreatic β-cell sensitivity to glucose levels Maternal hyperglycemia during pregnancy Maternal hyperglycemia caused by glucose infusions before or during labor Maternal drugs (anti-diabetic drugs, beta-blockers, antiarrhythmic drugs, interferon, valproic acid) Delayed feeding Excessive insulin administration during treatment of neonatal hyperglycemia
Patent neonatal hypoglycemia	Inborn metabolic errors (impaired gluconeogenesis): Carbohydrate metabolism errors Amino-acid metabolism errors Fatty acid metabolism errors Organic acidurias
	Congenital hyperinsulinism: ABCC8 and KCNJ11 mutations Glutamate dehydrogenase mutations Pancreatic β-cell hyperplasia Erythroblastosis Beckwith-Wiedemann syndrome Sotos syndrome
	Counter-regulatory hormone deficiency: Hypopituitarism Growth hormone deficiency Adrenal insufficiency Hypothyroidism
	Increased glucose requirement: Sepsis Hypothermia Hypoxia Polycythemia Cyanotic heart disease Exchange transfusion Intrauterine growth restriction Prematurity Mother starvation before labor

**Table 3 jcm-11-06020-t003:** Suggested laboratory investigations of the critical samples (taken during a hypoglycaemic episode) for differential diagnosis of neonatal hypoglycaemia.

Suggested Investigations in Differential Diagnosis
Complete blood count
Arterial blood gas
Blood glucose
Lactate
Pyruvate
Alanine
Glycerol
Ketone bodies
Plasma insulin
Free fatty acids
C-peptide
Plasma total and free carnitine, acylcarnitine profile
Ammonia
Electrolytes, blood urea nitrogen, creatinine
Uric acid
Growth hormone
Cortisol
Thyroid hormones
IGF-1
Galactosaemia screen
Ketones and organic acids in urine

**Table 4 jcm-11-06020-t004:** Levels of the biochemical parameters in the critical blood sample obtained during a hypoglycaemic episode.

Parameter	Level	Reference Range	Unit
Glucose	46	50–200	mg/dL
Insulin	183.5	2.6–24.9	µIU/mL
C-Peptide	18.61	1–4	ng/mL
Cortisol	19.0	6.2–19.4	µg/dL
Thyroid-stimulating hormone (TSH)	14.5	0.7–15.2	µIU/mL
Thyroxine (T4)	2.31	0.86–2.49	ng/dL
Growth hormone (GH)	22.4	1.18–27	ng/mL
Insulin-like growth factor (IGF-1)	47.22	0–26	ng/mL
Ammonia	45.7	21–95	µmol/L

**Table 5 jcm-11-06020-t005:** Comparison of the pre- and post-surgical exocrine and endocrine functions of the pancreas in our patient.

Parameter	Before Surgery	After Surgery	Reference Ranges	Unit
Glucose	46	105	50–160	mg/dL
Insulin	183.5	14.83	2.6–24.9	µIU/mL
C-Peptide	18.61	2.67	1.1–4.4	µg/L
Lipase	-	12	0–37	U/L
Elastase in faeces	-	432	>200	µg/g

## Data Availability

Not applicable.

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
