# Peer review of "Congenital Hyperinsulinaemic Hypoglycaemia—A Review and Case Presentation"

_jcm, 2022, doi:10.3390/jcm11206020_

Round 1

Reviewer 1 Report

The authors have done a great job of putting it all together as a extensive review that highlights the complications in diagnosis, treatment and management of congenital hyperinsulinaemic hypoglycaemia. Summarizing the available literature in the form of tables are highly commendable.  I have very few concerns and suggestions. 

1. Typo errors can be found throughout. 

2. It is highly recommended to include explanation for the figure-1 for better understanding of the concept.

3. In line 38: "channelopathies and metabolopathies"... Authors are suggested to explain the terms and then follow by "The common mutations are those affecting the KATP channel genes". 

4. How does somatostatin decreases insulin secretion by inhibiting the KATP channel?

5. Does diazoxide have any effect on Na+ channels or sodium reabsorption in kidneys since one of the side effects was hypertension?

6. Is there a role of inflammation/lymphatics in the pathogenesis of congenital hyperinsulinaemic hypoglycaemia?

Author Response

Response to Reviewer 1 comments:

Thank you very much for valuable remarks. The manuscript was improved according your’s suggestions

  1. The style of the article is British English. Therefore, we suggest to change the title from “Congenital hyperinsulinemic hypoglycemia – a review and case presentation” to ”Congenital hyperinsulinaemic hypoglycaemia – a review and case presentation”. We checked the article in search of typo errors but were not able to find any more – the Reviewer’s impression of typo errors might have come from the fact that the article is written in British English and not in American English (which is much more popular in scientific papers). Naturally, if the Editors prefer the American English style, we can revise our work according to such requirement. If any more typo errors are found, we kindly please to inform us.
  2. We added explanation for the Figure 1. It can be found under the figure.
  3. The terms have been explained
  4. It was our mistake. Somatostatin decreases insulin secretion by activating and opening KATP channels.
  5. Diazoxide acts like hypotensive agent but it influences sodium reabsorption in kidneys. It leads to fluid retention. We added this information to the section describing diazoxide. One of diazoxide side effects is pulmonary hypertension. It is thought that pulmonary hypertension is caused by the direct cardiotoxicity of diazoxide.
  6. We could not find any works concerning a role of inflammation/lymphatics in the pathogenesis of CHH.

Reviewer 2 Report

The manuscript is well written and the subject comprehensively covered. A few areas need further attention.

In figure 1 SCLA16A1 should be SLC16A1.

Considering the unresponsiveness of some ABCC8 genetic variants to diazoxide and considering the Recommendation of the Pediatric Endocrine Society (https://doi.org/10.1016/j.jpeds.2015.03.057) and fx. PMID: 17114887 the value of fast genetic screening before performing PET scanning should be addressed.

The necessity of performing intraoperative histology during surgery and the value of interoperative Ultrasound would be worth mentioning.

The genetic variant c.338CAT; p.Ala113Val is a known variant, PMID 16380471, 26092864, 27573238. The variant is listed in ClinVar under registration [VCV001338595.3]. Especially the variant is mentioned in author reference 11.

SUV values for the PET analysis should be mentioned (figure 3/line 343 p.12)

Considering the subject of the case report and the severity in the patient a relevant reference would be PMID: 31997554.

Author Response

Dear Reviewer 2 we improved the manuscript according your’s valuable advices

We corrected the mistake on Figure 1

We added the information about the value of fast genetic screening. It can be found at the end of section 3 (Diagnosis of HH)

We removed the information that this genetic variant was not listed in ClinVar.

The information about necessity of performing intraoperative histology has been added. We added the information about intraoperative ultrasound. Both can be found in the section „Surgical treatment”.

SUV values have been added (new Figure 3)

We added the summary and reference at the end of case report  
